# Adoptive NK Cell Therapy: A Promising Treatment Prospect for Metastatic Melanoma

**DOI:** 10.3390/cancers13184722

**Published:** 2021-09-21

**Authors:** Amanda A. van Vliet, Anna-Maria Georgoudaki, Monica Raimo, Tanja D. de Gruijl, Jan Spanholtz

**Affiliations:** 1Glycostem Therapeutics, Kloosterstraat 9, 5349 AB Oss, The Netherlands; anna@glycostem.com (A.-M.G.); monica@glycostem.com (M.R.); jan@glycostem.com (J.S.); 2Department of Medical Oncology, Cancer Center Amsterdam, Amsterdam UMC, Vrije Universiteit Amsterdam, De Boelelaan 1117, 1081 HV Amsterdam, The Netherlands; td.degruijl@amsterdamumc.nl

**Keywords:** Natural Killer cells, adoptive cell therapy, metastatic melanoma

## Abstract

**Simple Summary:**

The incidence of metastatic melanoma has been increasing over the past years with current therapies showing limited efficacy to cure the disease. Therefore, other options are being investigated, such as adoptive cell therapy (ACT) where activated immune cells are infused into a patient to attack melanoma. Natural killer (NK) cells are part of the innate immune system and extremely suitable for this kind of therapy since they show minimal toxicities in the clinical setting. In this review, we focus on current strategies for NK cell therapy and the development of new approaches that hold great promise for the treatment of advanced melanoma.

**Abstract:**

Adoptive cell therapy (ACT) represents a promising alternative approach for patients with treatment-resistant metastatic melanoma. Lately, tumor infiltrating lymphocyte (TIL) therapy and chimeric antigen receptor (CAR)-T cell therapy have shown improved clinical outcome, compared to conventional chemotherapy or immunotherapy. Nevertheless, they are limited by immune escape of the tumor, cytokine release syndrome, and manufacturing challenges of autologous therapies. Conversely, the clinical use of Natural Killer (NK) cells has demonstrated a favorable clinical safety profile with minimal toxicities, providing an encouraging treatment alternative. Unlike T cells, NK cells are activated, amongst other mechanisms, by the downregulation of HLA class I molecules, thereby overcoming the hurdle of tumor immune escape. However, impairment of NK cell function has been observed in melanoma patients, resulting in deteriorated natural defense. To overcome this limitation, “activated” autologous or allogeneic NK cells have been infused into melanoma patients in early clinical trials, showing encouraging clinical benefit. Furthermore, as several NK cell-based therapeutics are being developed for different cancers, an emerging variety of approaches to increase migration and infiltration of adoptively transferred NK cells towards solid tumors is under preclinical investigation. These developments point to adoptive NK cell therapy as a highly promising treatment for metastatic melanoma in the future.

## 1. Introduction

The incidence of melanoma has been increasing over the past years, especially among the elderly [1]. While early-stage melanoma can be cured by surgical resection, late-stage (metastatic) melanoma is commonly difficult to treat. For many years, the only two treatment options available were: Dacarbizine (DTIC), an alkylating chemotherapeutic agent which was approved by the US Food and Drug Administration (FDA) in May 1975, and IL-2, a cytokine which stimulates proliferation and function of T cells [2]. Since 2011, more therapies to treat metastatic melanoma have been approved, most notably immune checkpoint inhibitors and BRAF/MEK inhibitors. Despite these new therapies, 7180 out of 106,110 new melanoma cases will still result in death in the United States in 2021. Five-year survival rates are promising for patients diagnosed with early-stage melanoma (99%) but decrease to 27% with late-stage disease [3].

Melanoma is engaged in a complex crosstalk with the immune system, which influences the proliferation and differentiation of the tumor. Understanding the underlying mechanisms of interaction at the interface of melanoma and the immune system has been crucial for the development of new immunotherapy strategies. Current immunotherapies for advanced melanoma are designed to actively stimulate the patients’ own immune system by boosting T-cell activation (IL-2) or by blocking immune checkpoints [4,5]. Checkpoint inhibitors function by stimulating the immune response by targeting immunomodulatory molecules, such as CTLA-4 (ipilimumab) or PD-1 (nivolumab, pembrolizumab), while other therapeutics such as vemurafenib targets the V600E mutation of BRAF [6,7,8]. An alternative approach is to stimulate the patient’s immune system in a passive way by using adoptive cell therapy (ACT), such as autologous tumor infiltrating lymphocyte (TIL) therapy after ex vivo recovery and expansion. This approach seems to be very effective for patients who are diagnosed with late-stage melanoma, achieving an objective response rate of 49–72% when combined with a lymphodepleting regimen [9,10]. Although the number of ongoing clinical trials using adoptive T-cell therapy to treat various cancer types is high, no ACT, intended for the treatment of melanoma, has yet received FDA approval [11]. Most clinical trials are still in phase I or II. In addition, the field of chimeric antigen receptors (CAR) T-cell therapies is evolving, but these therapies can lead to severe toxicities caused by the cytokine release syndrome [12]. Thus, there is a pressing need for the development of efficient and safe targeted immunotherapies for the treatment of advanced melanoma.

In addition to the safety concerns about T cell therapies, it is also well known that tumors often develop strategies to escape T cell recognition. Antigens are presented to T cells in a human leukocyte antigen (HLA)-restricted manner, which triggers their activation and proliferation. By downregulating HLA class I molecules, tumor cells prevent T cell recognition and killing [13]. In contrast to T cells, NK cells are part of the innate immune system and can attack tumor cells without prior sensitization; their activity is dependent on the balance between signaling of activating and inhibiting receptors. Interestingly, lack of killer immunoglobulin-like receptors (KIR) engagement due to low or absent HLA-expression on the target site results in decreased inhibitory signaling input, which shifts the balance towards NK cell activation [14]. In vitro and in vivo (murine melanoma model) experiments have shown that melanoma cells, and even cancer stem cells, can shift the balance of NK cells to an activating status, leading to specific lysis of the tumor cells [15,16]. Due to their natural activity against tumor cells, and lack of off-target effects on healthy cells, NK cells are gaining increasing interest in cancer immunotherapy.

This review summarizes current approaches in NK cell-based ACT for metastatic melanoma, focusing on the biological rationale for the use of ACT and why NK cells are a highly promising immunotherapy for the treatment of advanced melanoma.

## 2. Dysregulation of the Immune System by Melanoma

Already in 1965, it was observed that certain melanomas spontaneously regressed [17]. Studies have shown that this regression was related to infiltration and expansion of cytotoxic T lymphocytes (CTL) in the tumor lesion leading to a curative immune response [18,19]. Moreover, it was found that melanoma patients with TILs have increased overall survival [20]. These TILs showed specific reactivity against autologous melanoma cells, and not against allogeneic tumor cells [21], and shared the same phenotype of activated CTLs [22]. Despite the anti-melanoma immune response, complete spontaneous regression of metastatic melanoma only occurs in 0.23% of the cases [23]. Thus, although melanoma is a very immunogenic tumor and induces an adaptive immune response, the patient’s immune system is usually incapable of clearing the tumor by itself. In these cases, the balance is shifted from tumor survival, where immune cells eliminate the tumor cells, to tumor immune escape resulting in tumor growth and spread [24]. 

Tumor immune escape can be driven by multiple mechanisms. Regulatory T cells (Tregs) play an essential role in maintaining homeostasis by suppressing excessive immune reactivity, thus maintaining self-tolerance. However, in melanoma patients the percentage of the Treg subset is increased from 2.24% to 7.93%, therefore leading to stronger immune suppression in the tumor microenvironment (TME) [25,26]. The lack of tumoricidal activity of TILs may be explained by high expression of immune checkpoints. TILs exhibit higher PD-1 expression than T cells from healthy tissue or peripheral blood, and typically have decreased levels of IFNγ [27]. Similarly, TIM-3 was found to be upregulated in melanoma-specific CD8^+^ T cells from peripheral blood of patients [28]. Another example of a checkpoint in the TME is the CD155-TIGIT pathway. CD155 is constitutively expressed on melanoma cells and activates T cells via the CD226 receptor. However, CD155 can also bind to TIGIT, which inhibits the function of CTLs. TIGIT is upregulated on TILs from melanoma patients and binds with great affinity to CD155, resulting in impaired function of the CTLs [29]. Evidence of immune involvement in melanoma development and progression has prompted the search for new immune therapies, such as ACT of ex vivo expanded autologous T cells with increased cytotoxic activity against melanoma.

## 3. Developments in ACT for Melanoma

The field of ACT for melanoma took flight in 1987, when an objective response rate of 34% was achieved with the use of TILs in combination with high dose IL-2 with or without cyclophosphamide [30]. Since then, multiple cell therapy strategies have been developed.

The responses observed in this first clinical study were short-lived, probably due to low persistence of the adoptively transferred T cells after infusion. To increase the in vivo persistence of T cells, new protocols for ACT were developed. A critical step in this process was the introduction of lymphodepleting regimens. Indeed, a significant correlation was found between persistence of adoptive T cell clones in peripheral blood and tumor regression upon lymphodepletion, suggesting that lymphodepletion is crucial to obtain a favorable clinical response [31]. Implementation of a lymphodepleting regimen resulted in objective response rates of 49–72% and 22% of patients with metastatic melanoma achieved a complete tumor regression [9].

Despite promising clinical response rates to TIL ACT, this therapy also has drawbacks. Before a patient can be treated with ACT, a rapid ex vivo expansion of tumor-specific TILs is required. A study by Dudley et al. showed that multiple TIL cultures from one melanoma biopsy are required to generate antigen-reactive TIL cultures. In 36% of the patients, ex vivo cultures failed to generate at least one tumor-specific TIL culture, and the treatment could not be administered [32]. Moreover, selection of TIL cultures and rapid expansion are time-consuming processes which pose significant challenges especially for patients with rapidly progressing cancers, such as melanoma. In addition, melanoma lesions and melanoma cell lines show down regulation of HLA class I, providing an immune escape mechanism from CD8^+^ T cell recognition and therefore also rendering ACT with ex vivo expanded T cells ineffective [13,33]. Other challenges that T cell ACT is faced with include the migration and infiltration of adoptively transferred T cells to tumor sites, the lack/downregulation of tumor-specific antigen expression and the immunosuppressive TME [34]. Due to these limitations, and while aiming to broaden the horizon in the field of cancer immunotherapies, researchers started to investigate other types of immune cells, such as NK cells.

## 4. Natural Killer Cells and Receptors Involved in Tumor Surveillance

Upon recognition of transformed cells, NK cells respond with the activation of a signaling cascade which leads to release of lytic granules or to direct, receptor-mediated induction of apoptosis. NK cells can also stimulate the immune response by releasing cytokines and chemokines. Classically, NK cells can be divided in two subsets based on their surface expression of CD56. CD56^bright^ NK cells produce high amounts of cytokines such as IFNγ to regulate other immune cells or directly affect tumor cell viability, whereas the CD56^dim^ NK cells are involved in direct cell-mediated cytotoxicity and show high CD16 expression levels. Unlike T cells, NK cells exert their functions via germline-encoded receptors which transduce activating or inhibitory signals. Thus, the activation status of NK cells is dependent on a sophisticated balance between the net signaling input of activating versus inhibitory receptors upon engagement of their respective ligands on tumor cells [35].

The most prominent activating receptors on NK cells are NKG2D (CD314) and DNAM-1 (CD226). These molecules are linked to an activating signaling pathway and play a major role in tumor cytotoxicity. NKG2D is a transmembrane receptor, belonging to the C-type lectin-like receptors, and can also be found on subsets of T cells [36]. The signaling domain of NKG2D is DNAX-activation protein 10 (DAP10), which upon activation is phosphorylated by the kinases of the Src family. Phosphorylation leads to binding to the P85 subunit of Phosphoinositide 3-kinase (PI3K) or to the adaptor molecule Growth factor receptor-bound protein 2 (Grb2), followed by recruitment of the downstream effector molecules SLP-76 and PLC-γ. Both P85 and Grb2 are required to stimulate calcium flux which leads to induction of exocytosis of lytic granules and cytotoxicity [37,38]. In contrast to most NK cell receptors, NKG2D signaling is not dependent on Zap70 or Syk kinases [39]. Its ligands include MHC class I chain-related molecules (MICA/B) and human cytomegalovirus ULl6-binding proteins (ULBP1-6) [40,41]. MICA/B is expressed in a broad range of tumor types including melanoma, while it is poorly expressed on the surface of healthy cells [42,43]. ULBPs are also expressed on a variety of tumor cell types, but in general low on melanoma cells [44]. Tumorigenesis induces upregulation of MICA/B and ULBPs via stress pathways [45]. DNAM-1 is a cell surface immunoglobulin type I glycoprotein and functions more as an adhesion molecule by synergizing with other activating receptors, such as 2B4. Its expression is not limited to NK cells, as DNAM-1 can also be found on the surface of T cells, a subset of B cells, monocytes, and platelets [46]. Known ligands are PVR (CD155) and Nectin-2 (CD112), which are expressed by a variety of different tumor types [47]. Upon binding to its ligand, LFA-1 co-localizes with DNAM-1, inducing phosphorylation of DNAM-1 and subsequently of the SH2-domain of SLP-76 [48]. This leads eventually, via PLC-γ, to calcium flux and secretion of cytokines.

The natural cytotoxicity receptors (NCRs), which consist of NKp30 (CD337), NKp44 (CD336) and NKp46 (CD335), were initially described to be activating receptors [49]. However, NKp30 and NKp44 can also be inhibiting upon binding to specific ligands [50,51]. The NCRs belong to the immunoglobulin superfamily. While NKp30 and NKp46 are constitutively expressed on NK cells, NKp44 is only expressed upon activation with IL-2. Differences are also observed in their downstream signaling. The associated adaptor proteins for NKp30 and NKp46 are FcεRIγ and/or CD3ζ chains, while for NKp44 it is DAP12 [52,53,54]. The proximal signaling pathway of the NCRs is similar and includes the phosphorylation of the associated immunoreceptor tyrosine-based activation motifs (ITAMs), recruitment of Syk/ZAP70, and activation of downstream signaling molecules PLC-γ, PI3K, and Vav1/2/3, leading to calcium flux and cytokine secretion [55]. Numerous ligands have recently been described for the NCRs and research is still ongoing to understand viral, bacterial, fungal or tumoral interactions. Here, we focus on the tumor associated ligands. B7-H6 and BAT-3 have been described as tumor cell-expressed ligands for NKp30 [56,57]. Interestingly, B7-H6 seems to be specific for tumor cells, as no expression was found on healthy cells [58]. BAT3 is a nuclear factor, but it can be released from tumor cells after heat shock, and binding to NKp30 leads to cytokine release [59]. Some intracellular ligands for NKp44 have been described, such as 21spe-MML5 and proliferating cell nuclear antigen (PCNA) [50,60]. During tumor transformation, these proteins can be transported to the surface of tumor cells, leading to NK cell activation upon binding to 21spe-MML5, whereas binding to PCNA inhibits NK cell effector function. Recently, it was discovered that NKp44 can also recognize extracellular ligands such as Nidogen-1 and a subset of HLA-DP molecules [61,62]. Although specific tumor ligands for NKp46 are not well described yet, it is known that metastases of melanoma do express ligands for NKp46, as was shown by using a NKp46-Fc staining [15]. All three NCRs can also bind to heparan sulfate proteoglycans (HSPGs), although to different molecules of this family [63].

NK cells are also equipped with receptors that recognize MHC class I molecules, such as NKG2A, NKG2C and the killer-cell immunoglobulin-like receptors (KIRs) [64]. NKG2A and NKG2C form heterodimers with CD94 and interact with the non-classical MHC-I molecule HLA-E [65]. NKG2A, an inhibitory receptor, is associated with an immunoreceptor tyrosine-based inhibition motif (ITIM), which is phosphorylated by Src family kinases after ligand binding. This leads to recruitment of SHP-1 and SHP-2, which can dephosphorylate the downstream signaling molecules of activating receptors such as Vav1 and SLP-76, and thereby inhibit NK cell activation [66,67]. In contrast, the activating receptor NKG2C is associated with an immunoreceptor tyrosine-based activation motif (ITAM) and contains the adaptor protein DAP12 [68]. Its downstream signal transduction is the same as for NKp44.

KIRs bind to the classical MHC-class I molecules and can also inhibit or activate NK cell-mediated killing. KIRs with a long intracellular tail contain an ITIM inhibitory domain, while KIRs with a short intracellular tail contain an ITAM activating domain. The only exception is KIR2DL4, which contains a single ITIM, but also carries a charged arginine residue in its transmembrane region, suggesting that it has both inhibiting and activating functions [69]. Almost all healthy cells express self-MHC-I molecules under physiological conditions, thus suppressing NK cell activity by engaging inhibitory KIRs. However, tumor or infected cells can downregulate MHC-I molecules to evade T cell recognition. The immune system has an elegant way of safeguarding against such immune escape through NK cell activation due to lack of inhibitory signaling through KIRs. This mechanism of action is termed the “missing-self hypothesis” [14]. This lack of inhibition by self-MHC molecules, which is a tumor escape mechanism from T cell recognition, results in attack by NK cells. Importantly, this release of inhibition alone is not sufficient to fully activate NK cells, since cells without MHC-I molecules are not attacked. Additional activating ligands on tumor cells are required for NK cells to reach a fully activated status [70].

The Fcγ receptor IIIa (FcγRIIIa), or CD16, is expressed on CD56^dim^ NK cells and is a member of the immunoglobulin superfamily. This receptor mediates antibody-dependent cellular cytotoxicity (ADCC); upon binding to the Fc region of an antibody, NK cells are activated to release cytokines and granules. The adaptor proteins of the CD16 receptor are FcεRIγ and/or CD3ζ chains [71]. Proximal signaling occurs via the classical ITAM pathway. In contrast to other NK cell receptors, CD16 does not require co-activation by other activating receptors to induce cytotoxicity or cytokine release [72].

The NKp80 receptor (KLRF1) is a type II transmembrane glycoprotein and can be found on NK and a subset of T cells [73]. The ligand for NKp80 is activation-induced C-type lectin (AICL) and is mainly found on hematopoietic cells, but it is also expressed on tumor cell lines [74]. NKp80 contains a hemi-ITAM and signals via Syk kinases [75].

The receptors 2B4 (CD244), CRACC (CD319) and NTB-A are all members of the signaling lymphocytic activation molecule (SLAM) family of receptors and function as co-receptors. These receptors are not NK cell-specific, as they can be found on many other immune cells. The ligand of 2B4 is CD48, which is expressed on hematological cells and some types of tumors. Binding to CD48 leads to phosphorylation of the immunoreceptor tyrosine-based switch motifs (ITSMs) but can lead to either a positive or negative signaling depending on which signaling molecules bind to which ITSM [76]. NTB-A and CRACC are homophilic receptors [77]. Receptor engagement leads to increased cytotoxicity, but its effect can be neutralized by the presence of inhibitory signals from other NK cells, to prevent fratricide [78].

## 5. The Possible Role of NK Cells in Immune Control of Melanoma

The skin contains a complex network of lymphocytes to defend the body against pathogens; the innate immune response is especially important in this first line of defense barrier. Healthy skin contains a large population of CD16-negative NK cells, which differs from the predominantly CD16-positive NK cell population found in the blood [79]. This population is mainly found in the dermis and expresses the chemokine receptor CCR8, indicative of a homing phenotype for non-inflamed skin [80]. These non-active NK cells have the potential to become active and then exert cytotoxic function towards stressed or infected cells, such as melanoma cells, as was shown by in vitro stimulation [79]. This suggests a role for NK cells in early tumor clearance in skin sites. More recently, it was also shown that NK cells are present in melanoma metastases and that this population of NK cells shows a different gene expression pattern than blood NK cells, reflecting a specialized functionality for cytotoxicity and chemokine/cytokine release [81]. Thus, melanoma can trigger NK cells to become active killers.

To achieve a positive NK cell response, it is required that activating ligands are highly expressed on the tumor cells. A large melanoma cell line panel was screened by Casado et al. for their ligand expression. The authors found that 85% of the cell lines expressed NKG2D ligands and 95% expressed DNAM-1 ligands, mainly MICA/B and PVR but not ULBPs or nectin-2 [82]. This suggests that the NKG2D-NKG2DL and DNAM-1-DNAM-1L axes play an important role in NK cell activation against melanoma, which is supported by the results of cytotoxicity blocking assays using antibodies inhibiting these receptor-ligand interactions [82]. In addition, expression of MICA/B was also found in both primary and metastatic melanoma patient samples [42]. Furthermore, ligands for NKp30, NKp44, and NKp46 were found to be expressed on melanoma cell lines [83,84].

### NK Cells as a Prognostic Factor

Interestingly, NK cells can also serve as a prognostic biomarker for melanoma progression. Although there is no overall difference in the number of circulating NK cells between healthy donors and melanoma patients, lower percentage of CD56^bright^ NK cells in melanoma patients correlates with shorter overall and progression-free survival [85]. The same correlation was observed for NKp46 on circulating NK cells in stage IV melanoma, where higher expression was associated with longer survival [86]. However, Pico de Coaña et al. reported that no correlation between NKp46 expression and overall survival was found, but that low expression of CD69 could be a predictive marker for better overall survival [87]. Using a different approach, analyzing RNA-seq data from bulk tumor samples, Cursons et al. used a NK cell signature to predict NK cell infiltration into melanoma and showed that this correlated with a better survival [88].

## 6. NK Cell Deficiency and Immunosuppression in Melanoma

Although NK cells were found to be active towards melanoma cells in vitro, multiple studies have shown impaired NK cell function in melanoma patients. As early as in 1980, Hersey et al. suggested that the cytotoxic activity of peripheral NK cells was inhibited in stage-I melanoma patients. Using a ^51^Cr release assay, they observed an increased cytotoxic capacity of peripheral blood NK cells after surgical removal of the localized melanoma compared to before removal [89]. This was supported by the finding that peripheral blood NK cells from stage IV melanoma patients showed decreased lysis of tumor cell lines compared to NK cells from healthy donors [90,91].

### 6.1. Downregulation of Activating Receptors on NK Cells

One of the tumor immune suppression mechanisms is the downregulation of activating receptors on NK cells. Pietra et al. showed that, upon co-culture with different melanoma cell lines, NK cells show downregulation of the activating receptors NKp30, NKp44 and NKG2D, leading to decreased cytolytic activity. This effect is due to tumor-mediated release of indoleamine-pyrrole 2,3-dioxygenase (IDO) and prostaglandin E2 (PGE_2_), two factors which control immune surveillance within the TME [92]. Downregulation of activating receptors was also observed by da Silva et al., who compared peripheral blood NK cells derived from melanoma patients versus healthy volunteers. In addition, they observed higher expression of the TIM-3 immune checkpoint on the surface of NK cells derived from melanoma patients, leading to impaired cytotoxic function [93].

### 6.2. Shedding or Downregulation of Activating Ligands on Melanoma Cells

Another immune escape mechanism is the shedding of ligands from melanoma cells, resulting in decreased engagement of activating receptors on NK cells in the TME. Shedding of ligands MICA, ULBP2, and B7-H6 has been observed in melanoma patients [94,95]. Blocking this phenomenon with an antibody that specifically targets the site of proteolytic cleavage, thus inhibiting the binding of matrix-metalloproteases (MMPs), on melanoma cells increases NK cell functionality giving rise to new approaches for combination therapy [95,96].

Since BRAF mutations are common in melanoma, BRAF-inhibitors are often used and have a direct effect on tumor regression. Unfortunately, drug resistance for such therapies develops very quickly [97]. As a consequence of the BRAF inhibitor treatment, activating ligands MICA, ULBP3 and PVR are decreased on melanoma cells, thus contributing to diminished NK cell activation and further support of immune escape [98].

### 6.3. The Suppressive Tumor Microenvironment

The TME of melanoma comprises immunosuppressive immune cells such as myeloid-derived suppressor cells (MDSC), T regulatory (Treg) cells and tumor-associated macrophages (TAM). These cells release factors such as PGE_2_, IDO, IL-10, Arginase-1, TGF-β and reactive oxygen species (ROS), which inhibit NK cell anti-tumor activity [99].

Not only do melanoma cells adapt to escape from the immune system, but the fibroblasts in the TME also play an important role in this process. Fibroblasts from melanoma patients secrete a higher level of active MMPs than fibroblasts from healthy controls. MMPs cause shedding of MICA/B from the surface of the tumor cells, resulting in decreased cytotoxic activity of NK cells [100]. In addition, melanoma-associated fibroblasts can also modulate NK cells directly by inhibiting cytokine-induced upregulation of NKp44, NKp30 and DNAM-1 [101].

More recently, it was shown that melanoma-derived exosomes negatively affect NK cell function, as they contain melanoma-associated antigens, MICA, and death ligands such as FASL and TRAIL. Co-culture of activated NK cells with melanoma-derived exosomes results in decreased levels of NKG2D [102].

An overview of the interplay between TME and melanoma is shown in Figure 1. Taken together, these data show that NK cells play an important role in melanoma immune surveillance, but NK cell function is often impaired both in the TME as well as in the periphery. Therefore, the use of adoptive NK cell transfer allows for multiple options for the treatment of melanoma patients. 

## 7. Clinical Exploration of Adoptive NK Cell Therapy for Melanoma

Safety and efficacy of adoptive NK cell therapy have been tested in multiple studies for the treatment of hematological malignancies (reviewed in [103]). However, the use of NK cells to treat solid tumors faces additional challenges, due to insufficient NK cell migration to and infiltration into the tumor. Preclinical studies are now focusing on improving these functions of NK cells for adoptive transfer. Current NK cell therapy approaches utilize autologous or allogeneic NK cells or NK cell lines.

### 7.1. Autologous NK Cell Therapy

In 2011, Parkhurst et al. used autologous NK cells to treat melanoma patients. In this study, seven patients with melanoma received in vitro-activated autologous NK cells after non-myeloablative, lymphodepleting chemotherapy. No objective clinical response was observed in the patients (see Table 1). Authors attributed this to the low expression of NKG2D on the persisting NK cells, which is necessary to exert cytotoxic function [104]. Another explanation is the self-tolerance of autologous NK cells, as described earlier. Tumor cells express HLA molecules that match with the KIR molecules on autologous NK cells, therefore resulting in no activation of the NK cells due to lack of alloreactivity.

### 7.2. Allogeneic NK Cell Therapy

To overcome inactivity due to self-tolerance, strategies where KIR-expression on allogeneic NK cells is mismatched to the HLA ligand expression on tumors were developed to achieve the highest NK cell activity. Initially, KIR-ligand mismatching was applied in a hematopoietic transplant setting [108]. Later, it was also applied in a study by Miller et al. where they used haploidentical allogeneic peripheral blood NK cells in a phase I clinical trial in 2005. The 10 metastatic melanoma patients enrolled in this trial received low dose cyclophosphamide/methylprednisolone followed by adoptive NK cell transfer. A stable disease state was observed in four subjects after the first infusion. However, after the second infusion, disease progression was observed [105] (see Table 1). In this same study, in vivo adoptive NK expansion was observed when patients were pre-treated with high dose cyclophosphamide/fludarabine, suggesting that results could perhaps be improved by pretreating melanoma patients with this high dose of lymphodepleting regimen. Other sources of allogeneic NK cell therapy, besides peripheral blood, are (stem cells from) umbilical cord blood and bone marrow, as well as induced pluripotent stem cells (iPSCs)(reviewed in [109]). 

### 7.3. NK Cell Line Therapy

A great advantage of using NK cell lines as a source for ACT is their unlimited supply. They can be expanded into larger numbers in a short period of culture. However, the NK cell lines need to be irradiated prior to infusion for safety reasons, which thereby limits their persistence and efficacy. Arai et al. (2008) used the NK cell line NK-92 in a phase I clinical trial: only one melanoma patient was included and showed minor response after infusion (see Table 1). However, this patient died 255 days post-NK cell infusion because of disease progression [107]. This trial did not include pre-treatment with lymphodepleting regimen and therefore three doses of NK-92 cells were infused in a short period (5 days) to achieve high dose of NK-92 before a patient T cell immune response could occur to allow for NK-92 cytotoxicity.

### 7.4. Ongoing Clinical Trials

Besides ongoing clinical trials with NK cells as monotherapy, the focus is also expanding into combining NK cell therapy with other agents. Combination therapy helps to increase NK cell therapy efficacy, thereby reducing the burden of high dose infusions or repetitions. Notably, the NCT00720785 trial uses autologous NK cells in combination with bortezomib, a proteasome inhibitor that has been shown to increase NK cell mediated cytotoxicity via upregulation of the FAS and TRAIL receptors [110]. Other trials include combinations of NK cells with immune checkpoint inhibitors such as nivolumab, targeting PD-1 or atezolizumab, targeting PD-L1 (NCT03841110). 

Table 1 provides an overview of previous and ongoing clinical trials with NK cells for the treatment of melanoma and as can be observed, only a limited number of trials is registered including published results from only three trials (as being discussed above). Moreover, all are using a different NK cell source. Therefore, more results need to be generated in the future to be able to conclude about the efficacy of NK cell ACT for melanoma and what the effect is of a lymphodepleting regimen. 

## 8. Future Developments in NK Cell Therapy for Melanoma

With our increasing understanding of the mechanisms underlying NK function, as well as development and progression of melanoma, advanced molecular technologies can be applied to design more specific (or even personalized) NK cell therapies for melanoma. Adoptive NK cell therapy will likely not represent a stand-alone therapy, but its efficacy could be enhanced either in combination with other therapies or by genetic engineering. Examples of such strategies are discussed below and summarized in Figure 2. 

### 8.1. Migration to the Tumor Site

Successful anti-cancer therapy requires migration of NK cells to the tumor site. Chemokines are major players driving this migration. A promising strategy is to target the autophagy pathway by silencing Beclin1 (BECN1), leading to improved NK cell infiltration into the tumor because of increased levels of the chemokine CCL5. Although these findings were observed in mouse models, the same study found a positive correlation between survival and high expression of CCL5 in melanoma patients, confirming that this chemokine plays an important role in the human setting [111]. In addition, the CXCR3-CXCL10 axis was found to be important for NK cell migration to melanoma [112]. Thus, stimulating the tumor to secrete higher levels of CXCL10 could be a successful way to enhance the efficacy of adoptive NK cell therapy. Another approach is to increase chemokine receptor expression on the NK cells. Since many solid tumor types, including melanoma, have high expression of IL-8 in the TME, Yang Ng et al. overexpressed CXCR1, the receptor for IL-8, in ex vivo expanded NK cells and performed in vivo mouse studies which showed increased migration of the modified NK cells to the tumor [113].

### 8.2. Inhibition of Immunosuppression in the Tumor Microenvironment 

Immunosuppression in the TME is often caused by factors that are either released by melanoma cells or by immune cells itself [99]. Clinical attempts to inhibit these factors are underway, but trials in which adoptive NK cell therapy is used in combination are very limited. However, preclinical data showed that the TGF-β inhibitor LY2157299, in combination with adoptive NK cell therapy, resulted in better tumor suppression than LY2157299 or NK cells alone in a mouse model for metastatic colon cancer [114].

### 8.3. Killer Engagers

As mentioned earlier, tumors escape from NK cell immune surveillance by shedding the MICA/B ligands. This can be prevented by using a monoclonal antibody which specifically recognizes the key epitopes on the MICA and MICB α3 domain (which is essential to initiate shedding), without interfering with the binding of MICA/B to NKG2D. In addition, the Fc part of the antibody contributes to cytotoxic efficacy by engaging the CD16 receptor on NK cells leading to ADCC [96]. Killing engagers do not need to bind to the NK cell site via Fc-CD16 but could also bind directly to an NK cell receptor and a tumor antigen to engage killing. Pekar et al. engineered a bispecific engager that recognizes NKp30 on the NK cells and EGFR on the tumor cells, using an antigen binding fragment (Fab) arm derived from cetuximab, and showed that NKp30 engagement was sufficient to induce NK cell cytotoxicity. However, when adding a human Fc-competent IgG1 fragment to the engager, it resulted in an almost 10-fold higher killing capacity [115]. Another example is a trifunctional NK-cell engager that targets NKp46 and CD16 on the NK cells and a tumor antigen on the tumor cell, showing better efficacy than standard monoclonal antibodies, such as cetuximab or rituximab [116]. Combinations of antibodies or engagers with NK cell therapy would be a valuable approach to achieve greater anti-tumor efficacies.

### 8.4. NK Cell Receptor Modification

While tumor-infiltrating and peripheral NK cells can downregulate activating receptors, adoptively transferred NK cells with overexpression of those receptors can be used to overcome the resistance of an immunosuppressive TME. It has been shown that overexpression of NKG2D and DNAM-1 in NK-92 cells results in especially higher degranulation against melanoma cell lines A375, Mewo and SK-MEL-28 compared to wild type NK-92 [117].

### 8.5. CAR-NK Cells

To increase recognition of tumor cells, NK cells can be genetically modified to express a chimeric antigen receptor (CAR), which specifically recognizes tumor-associated antigens. The development of CAR-T cell therapy for melanoma is rapidly progressing, but the development of CAR-NK cells is lagging as it is currently limited to the pre-clinical phase. Melanoma-specific antigens that are currently under investigation in clinical trials using CAR-T cells include c-MET, CD70, GD2 and VEGFR2 [118]. Another prominent melanoma-specific tumor antigen is the high molecular weight melanoma-associated antigen (HMW-MAA), also called chondroitin sulphate proteoglycan 4 (CSPG4), which is present in more than 90% of melanomas [119]. By electroporating NKT cells with RNA encoding the receptor that recognizes HMW-MAA, Simon et al. generated functional CAR-NKT cells which expressed lower levels of cytokines than CD8^+^ T cells, thus reducing the risk of cytokine release syndrome [120]. In another approach, a lentivirus-based transduction platform was used to create a universal CAR (UniCAR) NK-92 cell line recognizing the E5B9 epitope. In combination with a switchable tumor-specific target module that consists of the E5B9 epitope fused with an antigen-binding moiety, UniCAR NK-92 can be used against multiple targets [121,122]. Implementing a GD2-specific target module, Mitwasi et al. showed specific lysis of GD2 expressing melanoma cells in vitro [123]. Moreover, CD276 (B7-H3), which is highly expressed on melanoma, is efficiently targeted by CAR NK-92 cells when tested in 2D or 3D in vitro settings [124].

### 8.6. TCR-NK Cells

While CARs can only recognize extracellular antigens, T cell receptors (TCRs) are able to recognize both extracellular and intracellular antigens via antigen processing and peptide/MHC presentation. Although NK cells do not express the TCR complex subunits (except for CD3ζ), they do express the downstream signaling molecules of the TCR complex [125]. This makes it possible to genetically modify NK cells to express a TCR. Parlar et al. demonstrated a proof-of-principle study with genetically modified NK cell lines expressing a TCR specific for the HLA-A2-restricted tyrosinase-derived “YMDGTMSQV” epitope. Besides functional TCR-NK cell signaling, an increase in NKp30 and NKp46 receptor expression was observed, since they were CD3ζ-coupled, suggesting that the NK cells retain the natural cytotoxic function. Zhang et al. performed genetic modification of NK cells specifically targeting a melanoma-associated antigen. They used a TCR-like antibody, GPA7, which recognizes gp100/human leukocyte antigen (HLA)-A2 complex and fused this together with the intracellular CD3ζ domain. Remarkably, they observed specific lysis of melanoma cells via TCR signaling [126]. Mensali et al. showed that NK-92 cells transduced to express both the TCR and the CD3 complex gained similar effector functions as T cells when the TCR was stimulated [127].

### 8.7. Immune Checkpoint Inhibitors

Recently, researchers have shown rising interest in the immune checkpoint receptor NKG2A, which is a promising target to increase cytotoxic efficacy of NK cells [128,129]. It has also been shown that melanoma cells have higher HLA-E expression levels than healthy melanocytes, and that this expression leads to protection from lysis by NK cells, as was confirmed by NKG2A blocking experiments [130]. Inhibiting NKG2A could release the breaks to gain higher efficacy of NK cell therapy. A similar effect could be achieved by blocking inhibitory KIR receptors. This approach has already been tested in several clinical trials using the anti-KIR antibody IPH2101 for acute myeloid leukemia (NCT01687387) or multiple myeloma (NCT00552396) as monotherapy. Although no clinical response has been shown yet, it still could be of great benefit in combination with adoptive NK cell therapy.

### 8.8. Use of Nanoparticles

Nanoparticles can be deployed to deliver therapeutic reagents such as antibodies, cytokines and genes, and are a promising combination strategy to enhance NK cell functionality. Multiple strategies are under development including inhibition of the immunosuppressive TME, improving migration and infiltration, and augmenting receptor-ligand activation response (reviewed in [131]). An example of TME modulation is the use of nanoscale liposomal polymeric gels to co-deliver a TGF-β inhibitor and IL-2 to the TME, resulting in decreased tumor growth and increased numbers of NK cells in the tumor of a B16 mouse melanoma model [132].

## 9. Concluding Remarks

Due to its high immunogenicity, melanoma is extremely suitable for immune therapy. Over the years, multiple strategies have been explored to stimulate the adaptive or innate immune system in the fight against melanoma. It has become clear that this disease is very heterogeneous and needs a specialized treatment, adapted to the individual phenotype of the patient’s tumor. Increasing knowledge contributes to the understanding of the complex crosstalk between melanoma and immune cells, leading to the development of novel therapeutics of enhanced efficacy.

In recent years, NK cells have become part of the immune arsenal in the fight against metastatic melanoma. Preclinical evidence of the promising potential of NK cells as ACT for advanced melanoma is growing and deeper understanding of the suppressive effect of tumor cells on NK cells informs the design of new, more effective therapeutics.

The low off-target toxicity and natural cytotoxic capacity of NK cells against tumor cells make them excellent candidates for immune cell therapy, as confirmed by multiple clinical trials. Besides the advantageous low incidence of graft-versus-host disease, the potential of NK cells to become an off-the-shelf product supports widespread clinical use. Currently, the number of ongoing clinical trials using NK cell therapy for advanced melanoma is limited but will hopefully grow in the coming years as we gain more knowledge about the potency of NK cells. An effective strategy to make progress in the clinical development of NK therapies for melanoma could be combination of NK cell therapy with other approved therapeutic that manipulate the NK cells, the tumor and its microenvironment or the host immune system to achieve greater in vivo synergizing effects. NK cell therapy will likely not represent a monotherapy, but different combination strategies will be devised based on advanced tumor profiling technologies, which will allow sufficient degree of personalization to best address the individual tumor phenotype. Research will continue to identify suitable new combination therapies or engineering approaches to enhance NK cell efficacy with the ultimate goal of achieving curative anti-tumor responses with low toxicities in solid tumor cancer immunotherapy.

## Figures and Tables

**Figure 1 cancers-13-04722-f001:**
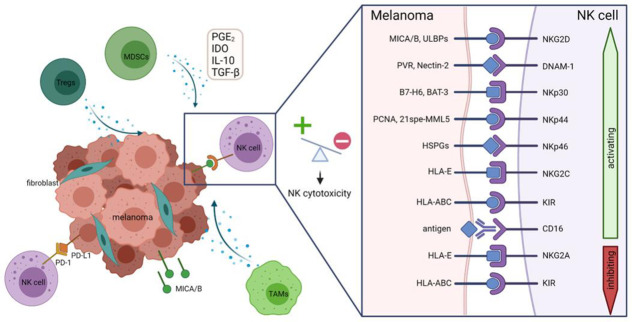
The TME shapes the activity of NK cells in melanoma. Different immunosuppressive immune cells such as Tregs, MDSCs and TAMs have a direct effect on the activity of NK cells by releasing factors such as PGE_2_, IDO, IL-10 and TGF-β. Even fibroblasts and the melanoma cells can release inhibiting factors. These factors can down or upregulate specific NK cell receptors or cause shedding of ligands, such as MICA/B on the melanoma site, leading to a shifted balance of signaling from activating to inhibiting status. This can result in NK cell deficiency. Ligand immune checkpoint expression on melanoma such as PD-L1 also influences NK cell activity.

**Figure 2 cancers-13-04722-f002:**
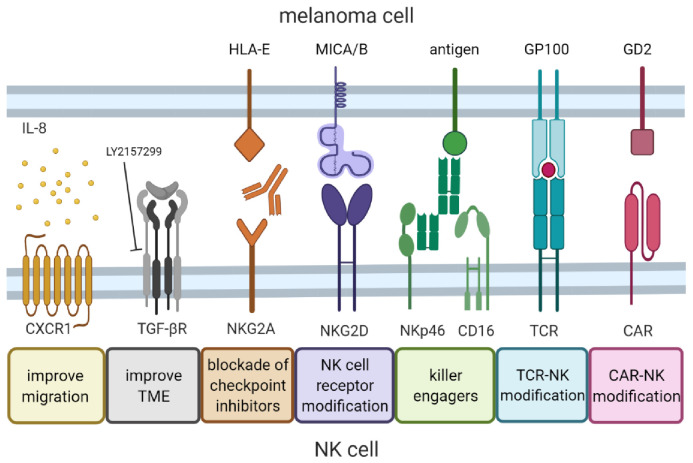
Strategies to improve NK cell therapy for melanoma. Different approaches can be implemented to improve NK cell therapy for melanoma. NK cells can be redirected to the tumor site by increasing migration via upregulation of chemokine receptor CXCR1. At tumor site, the immunosuppressive effect of the TME can be neutralized by inhibitors targeting immunosuppressive factors, such as TGF-β. Cytotoxicity can be improved by combination therapy, such as blocking checkpoint inhibitors (i.e., NKG2A) or using killer engagers such as bi/tri-specific antibodies. Another approach to improve cytotoxicity utilizes genetic modification of NK cells to overexpress activating receptors such as NKG2D, or arming NK cells with TCRs or CARs targeting tumor-associated antigens such as GP100 or GD2.

**Table 1 cancers-13-04722-t001:** Overview of clinical trials using NK cells for the treatment of melanoma.

NK Cell Source	Study (NCT Number/Reference) and Clinical Phase	Study Title	Start Date-End Date	Patient Inclusion	Cell Culture Method	Infusion Dose	Pre-Treatment	Outcome	Academic/Industry
Autologous NK cells	Parkhurst et al. [104], NCT00328861, phase II	Natural Killer Cells Plus IL-2 Following Chemotherapy to Treat Advanced Melanoma or Kidney Cancer	May 2006, April 2009	7, progressive stage IV melanoma	3-week culture period of CD3+ depleted cells using irradiated PBMCs as feeder cells	4.7 × 10^10^ (±2.1 × 10^10^) cells, +high dose I.V. IL-2 (720,000 IU/kg/dose)	2 days of cyclophosphamide (60 mg/kg) followed by 5 days of fludarabine (25 mg/m^2^)	no objective clinical responses	National Cancer Institute, Bethesda, Maryland
	NCT00720785, phase I	Natural Killer Cells and Bortezomib to Treat Cancer	Recruiting in Sep 2020	Malignant melanoma	Ex vivo expanded NK cells after apheresis	unknown	unknown	NA	National Cancer Institute, Bethesda, Maryland
Allogeneic NK cells	Miller et al. [105], phase I	-	2005	10, Metastatic melanoma	Depletion of CD3+ cells from PBMCs of haploidentical donors, NK cells were cultured overnight in medium containing IL-2	1 × 10^5^, 1 × 10^6^, 1 × 10^7^, or 2 × 10^7^ cells/kg of recipient body weight, +IL-2 injection for 14 days (1.75 × 10^6^ IU/m^2^)	48 hours prior to infusion: 750 mg/m^2^ intravenous cyclophosphamide and 1000 mg/m^2^ methylprednisolone	4 patients with stable disease. After second infusion, disease progression was found within 4 to 6 weeks	University of Minnesota Cancer Center, Minneapolis
	NCT00846833, phase I/II	Haploidentical NK Cell Infusion in Malignant Melanoma	Feb 2009–April 2012	Confirmed metastatic or relapsed malignant melanoma and who received prior chemotherapy or immunotherapy	Depletion of CD3+ cells from PBMCs of haploidentical donors	unknown	Cyclophosphamide	unknown	Seoul National University Hospital Seoul, Korea
	NCT03420963, phase I	Donor Natural Killer Cells, Cyclophosphamide, and Etoposide in Treating Children and Young Adults With Relapsed or Refractory Solid Tumors	Feb 2018–recruiting	Recurrent Cutaneous Melanoma, Refractory Cutaneous Melanoma	Cord blood derived expanded allogeneic NK cells	unknown	Cyclophosphamide and etoposide	NA	M. D. Anderson Cancer CenterHouston, Texas, United States
	NCT03841110, phase I	FT500 as Monotherapy and in Combination With Immune Checkpoint Inhibitors in Subjects With Advanced Solid Tumors	Feb 2019–recruiting	Melanoma	NK cell derived from a clonal master iPSC line	3 infusions, once weekly	Fludarabine/ Cyclophosphamide	NA	Fate therapeutics, San Diego, United states
	NCT03319459, phase I	FATE-NK100 as Monotherapy and in Combination With Monoclonal Antibody in Subjects With Advanced Solid Tumors	Jan 2018–recruiting	Melanoma	Allogeneic donor derived NK cells	unknown	unknown	NA	Fate therapeutics, San Diego, United states
	NCT03007823, phase I/II	High-activity Natural Killer Immunotherapy for Small Metastases of Melanoma	Dec 2016–Jun 2019	Small metastasis of melanoma	14 days of culture of allogeneic NK cells using a human NK cell in vitro culture kit (HANK bioengineering) [106]	8–10 ×10^9^ cells per infusion, 3 times infusion	unknown	unknown	Fuda Cancer Hospital, Guangzhou, China
NK-92 cell line	Arai et al. [107], phase I	-	April 2002–June 2004	malignant melanoma refractory with failed standard therapy	3 weeks of ex vivo culture of NK-92 cell line with IL-2. Irradiation before infusion	3 × 10^9^ cells/m^2^	Diphenhydramine prior to infusion	Minor response	Stanford University, Stanford, California, United States

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
