# Peer review of "Adoptive NK Cell Therapy: A Promising Treatment Prospect for Metastatic Melanoma"

_cancers, 2021, doi:10.3390/cancers13184722_

Round 1

Reviewer 1 Report

In this review, authors the authors have systematically put together the recent reports and clinical trials on NK cell therapy for metastatic melanoma. This review covers many different and relevant aspects of the NK cell therapy and its potential from recent literature. I have only a few minor suggestions as mentioned below.

Authors can add few lines on the aspect of nanoparticle mediated delivery of the NK cell therapy attempts (PMID: 33918941), and other strategies (PMID: 32411806).

Author Response

Authors can add few lines on the aspect of nanoparticle mediated delivery of the NK cell therapy attempts (PMID: 33918941), and other strategies (PMID: 32411806).

Dear reviewer,

Thank you for your feedback. The suggestion of commenting on the use of nanoparticles in combination with NK cell therapy is added to section 8 of the manuscript as a paragraph “use of nanoparticles”.  In this article, we tried to focus on NK cell therapy strategies that have already shown some pre-clinical results for the treatment of melanoma. Therefore, not all possible NK cell therapy strategies are discussed. However, we think that the most important strategies are covered.

Reviewer 2 Report

The authors have reviewed a very interesting topic in this manuscript. The manuscript is well written with a clear language. The review is well organized and comprehensive in its description of NK cells and the use of NK cells for adoptive transfer therapy. A few comments for the authors are listed below.

  • Correct the “lympho-depleting” at line 59 to lymphodepleting as written elsewhere in the manuscript.
  • The section about ACT therapy in combination with lyphodepleting agents (line 55-62) highlights this as a promising strategy and points to a number of clinical trials, although no treatment is FDA approved. It would be helpful to the reader to know if these trials are in early or late stage.
  • The manuscript would benefit from including additional illustrative figures.
    • A figure showing how the melanoma cross-talks with the immune system and how this influence on the tumor, control or escape is suggested (fits with sections 1 and 2)
    • A figure supporting section 4 that illustrates the various activating and inhibitory receptors on NK cells. Possibly also pointing to their (known and/or proposed) ligands. I enjoyed the details of this section describing the function of the NK cell receptors and their ligands and cellular expression. A figure here would also support later sections of the manuscript.
  • Suggest to include a brief description of which cells express MICA/B and how MICA/B expression is induced (maybe also ULBP1-6). Line 166/167.
  • The “ongoing clinical trials” section starting at line 374 could be slightly more detailed as to which immune checkpoint inhibitors or tumor-targeting mAbs are used in combination with NK cells and the rationale behind this.
  • Space between “receptors” and the ref at line 378.
  • Where suitable describing clinical trials in the text, please refer to Table 1 for details.
  • The authors could be more detailed on soluble, protein engagers in section 8. In the “Enhancement of ADCC” part an Fc containing engager is described. However, several engagers are explored that do not contain an Fc part and therefore do not mediate killing through ADCC, but rather by directly binding CD16 or other NK cell receptors and a tumor antigen, thereby bridging the two cells (eg BiKEs, TRiKES). I am aware that this is a bit to the side of the focus of this review, but the strategy is interesting and could be promising in combination with cell transfer as other strategies outlined in the manuscript.
  • Found the sentence “Mensali et al showed …” (line 468-471) to be unclear (implement and add TCR and CD3?). Would suggest rephrasing to “Mensali et al showed that NK-92 cells transduced to express both TCR and the CD3 complex resulted in….” or something similar.

Author Response

 Dear Reviewer,

Thank you for your helpful feedback. Please find below the responses to your suggestions.

  • Correct the “lympho-depleting” at line 59 to lymphodepleting as written elsewhere in the manuscript.

Done

  • The section about ACT therapy in combination with lymphodepleting agents (line 55-62) highlights this as a promising strategy and points to a number of clinical trials, although no treatment is FDA approved. It would be helpful to the reader to know if these trials are in early or late stage.

Most of the trials are in phase I or phase II and this is now added to the manuscript (line 62).

  • The manuscript would benefit from including additional illustrative figures.
    • A figure showing how the melanoma cross-talks with the immune system and how this influence on the tumor, control or escape is suggested (fits with sections 1 and 2)
    • A figure supporting section 4 that illustrates the various activating and inhibitory receptors on NK cells. Possibly also pointing to their (known and/or proposed) ligands. I enjoyed the details of this section describing the function of the NK cell receptors and their ligands and cellular expression. A figure here would also support later sections of the manuscript.

The authors would like to thank the reviewer for proposing detailed suggestions for illustrations that would enhance the understanding of the various mentioned sections by the reader. We would like to highlight that the focus of the review is on NK cells and that section 1 and 2 provide a general overview of the status of ACT and melanoma. We would therefore see it more fitting to prepare a figure that better describes the later sections to enhance the understanding of the complex nature of NK cell interactions in the TME. We added a new figure that incorporates the broad view of the TME in melanoma and also provides insight into the specific molecular receptor-ligand interactions of NK cells and melanoma in the TME.

  • Suggest to include a brief description of which cells express MICA/B and how MICA/B expression is induced (maybe also ULBP1-6). Line 166/167.

A short description about the expression and induction is added in line 168-172.

  • The “ongoing clinical trials” section starting at line 374 could be slightly more detailed as to which immune checkpoint inhibitors or tumor-targeting mAbs are used in combination with NK cells and the rationale behind this.

Rows 395-397 have been modified according to the reviewer’s suggestion to address the rationale for NK cell combination therapies. The details of the mAbs are added to this section (line 400-401).

  • Space between “receptors” and the ref at line 378.

Done

  • Where suitable describing clinical trials in the text, please refer to Table 1 for details.

Included, see line 361,375,388.

  • The authors could be more detailed on soluble, protein engagers in section 8. In the “Enhancement of ADCC” part an Fc containing engager is described. However, several engagers are explored that do not contain an Fc part and therefore do not mediate killing through ADCC, but rather by directly binding CD16 or other NK cell receptors and a tumor antigen, thereby bridging the two cells (eg BiKEs, TRiKES). I am aware that this is a bit to the side of the focus of this review, but the strategy is interesting and could be promising in combination with cell transfer as other strategies outlined in the manuscript.

The section “Enhancement of ADCC” is renamed as “Killer engagers”. This section is extended to better explain the different developments of antibodies that improve ADCC and engagers that are developed to also target other NK cell receptors. This strategy of directly targeting NK cell receptors is indeed very interesting and circumvents problems with NK cell subsets that express low CD16.

  • Found the sentence “Mensali et al showed …” (line 468-471) to be unclear (implement and add TCR and CD3?). Would suggest rephrasing to “Mensali et al showed that NK-92 cells transduced to express both TCR and the CD3 complex resulted in….” or something similar.

This sentence is changed according to the reviewer’s suggestion.

Reviewer 3 Report

I was glad to read the review describes the role of NK based ACTs in the treatment of skin cancer. All paragraphs are well written and compact. All major topics are touched, relevant pre-clinical and human trials are cited. 

Reviewer has a minor suggestion:

I think one extra figure representing the major molecular interactions that limits (or helps) NK cell action in  a tumor stroma would add an extra taste and beauty to the manuscript.

Author Response

Dear Reviewer,

Thank you for this great suggestion. We added a new figure which shows the overview of the TME and what effect this has on NK-tumor interaction (figure 1).

Reviewer 4 Report

This is a review article concerning NK-cell based therapy of cancer. Authors themselves are developing a kind of NK-cell immunotherapy for different cancer so that they have a personal experience with this topic and are aware about many hurdles connected with the cell-based therapy.

I have some suggestions how to improve the quality of the paper.

  1. The part 4 describing NK receptors and their function is too long and bring text-book-based informations. I recommend to shorten this part to main relevant facts connected with NK cell immunotherapy.
  2.  Part 6 ends by the statement: "Taken together, these data show that NK cells play an important role in melanoma  immune surveillance, but NK cell function is often impaired both in the TME as well as in  the periphery. Therefore, adoptive cell transfer of NK cells is a very promising treatment option for melanoma patients". However, in the following part 7 there is a description of clinical trials with NK cell therapy in melanoma which are rather disappointing. So I recommend to diminish the statement by a conditional like "...there is still a rationale to develop adoptive NK- cell transfer... 
  3.  In the part 7, I recommend to critically discuss more the reasons of the failure of described trials (dose, frequency, persistence of transferred cells,  tumor burden)
  4.  I recommend to explain how allogenic NK cell might work in a immunocompetent recipient.  Lymphodepletion give very short space and time for persistence of transferred NK cells.
  5. In part 8, I would recommend to bring some personal and critical view to the described strategies how to improve NK-cell based therapy. There are endless theoretical possibilities. In drug development, there is no way to test all of them.  Which of these approach the authors consider as the most promising way and why?   

Author Response

Dear Reviewer,

Thank you for your kind feedback. Please see below the response to your suggestions.

  1. The part 4 describing NK receptors and their function is too long and bring text-book-based informations. I recommend to shorten this part to main relevant facts connected with NK cell immunotherapy.

The authors respectfully disagree and prefer to leave this part in the manuscript, as two of the reviewers gave different opinions about this specific part and we strongly feel that it provides helpful information for readers who have less background in NK cells.

2.  Part 6 ends by the statement: "Taken together, these data show that NK cells play an important role in melanoma  immune surveillance, but NK cell function is often impaired both in the TME as well as in  the periphery. Therefore, adoptive cell transfer of NK cells is a very promising treatment option for melanoma patients". However, in the following part 7 there is a description of clinical trials with NK cell therapy in melanoma which are rather disappointing. So I recommend to diminish the statement by a conditional like "...there is still a rationale to develop adoptive NK- cell transfer... 

The statement in part 6 has been changed to “the use of adoptive cell transfer of NK cells allows for multiple options for the treatment of melanoma patients” to make a broader statement. It is true that the results of the clinical trials are not that promising yet, but a reason is also that the number of clinical trials is very low. A clear conclusion can therefore not be drawn yet. An extra paragraph is added to part 7 to discuss this (lines 402-407).

  1. In the part 7, I recommend to critically discuss more the reasons of the failure of described trials (dose, frequency, persistence of transferred cells,  tumor burden)

More details of the trials that are being discussed are added to part 7. Since the number of enrolled melanoma patients are so low, it is difficult to discuss the reason of failure.

4. I recommend to explain how allogenic NK cell might work in a immunocompetent recipient.  Lymphodepletion give very short space and time for persistence of transferred NK cells.

A comment about not using a lymphodepleting regimen is added (line 390-393). The trial that is described here did not use a lymphodepleting regimen. Unfortunately, only one melanoma patient was included in this trial, so strong conclusions can not be made. Even using low dose of lymphodepleting regimen did not lead to an expansion of adoptively transferred NK cells as is discussed in lines 375-378. So, from the current results, it is not possible to conclude of lymphodepleting regimen is required to obtain an clinical response with NK ACT for the treatment of melanoma.

5. In part 8, I would recommend to bring some personal and critical view to the described strategies how to improve NK-cell based therapy. There are endless theoretical possibilities. In drug development, there is no way to test all of them.  Which of these approach the authors consider as the most promising way and why?   

A short suggestion of how to move forward with NK cell therapy for melanoma is added to part 9 (lines 552-561).